# Genomic heterogeneity of multiple synchronous lung cancer

Yu Liu[1,*], Jianjun Zhang[2,3,*], Lin Li[1,4,*], Guangliang Yin[5,*], Jianhua Zhang[6,*], Shan Zheng[4], Hannah Cheung[2], Ning Wu[7], Ning Lu[4], Xizeng Mao[6], Longhai Yang[8], Jiexin Zhang[9], Li Zhang[7], Sahil Seth[6], Huang Chen[1], Xingzhi Song[6], Kan Liu[7], Yongqiang Xie[4], Lina Zhou[7], Chuanduo Zhao[8], Naijun Han[1], Wenting Chen[1], Susu Zhang[1], Longyun Chen[5], Wenjun Cai[5], Lin Li[5], Miaozhong Shen[5], Ningzhi Xu[10], Shujun Cheng[1], Huanming Yang[5], J. Jack Lee[11], Arlene Correa[12], Junya Fujimoto[13], Carmen Behrens[13], Chi-Wan Chow[13], William N. William[3], John V. Heymach[3], Waun Ki Hong[3], Stephen Swisher[12], Ignacio I. Wistuba[13], Jun Wang[5,14,**], Dongmei Lin[4,**], Xiangyang Liu[8,**], P. Andrew Futreal[2,15,**] & Yanning Gao[1,**]

Multiple synchronous lung cancers (MSLCs) present a clinical dilemma as to whether individual tumours represent intrapulmonary metastases or independent tumours. In this study we analyse genomic profiles of 15 lung adenocarcinomas and one regional lymph node metastasis from 6 patients with MSLC. All 15 lung tumours demonstrate distinct genomic profiles, suggesting all are independent primary tumours, which are consistent with comprehensive histopathological assessment in 5 of the 6 patients. Lung tumours of the same individuals are no more similar to each other than are lung adenocarcinomas of different patients from TCGA cohort matched for tumour size and smoking status. Several known cancer-associated genes have different mutations in different tumours from the same patients. These findings suggest that in the context of identical constitutional genetic background and environmental exposure, different lung cancers in the same individual may have distinct genomic profiles and can be driven by distinct molecular events.

[1] State Key Laboratory of Molecular Oncology, Department of Etiology and Carcinogenesis, Cancer Institute and Hospital, Peking Union Medical College and Chinese Academy of Medical Sciences, Beijing 100021, People's Republic of China. [2] Department of Genomic Medicine, The University of Texas MD Anderson Cancer Center, 1515 Holcombe Boulevard, Houston, Texas 77030, USA. [3] Department of Thoracic/Head & Neck Medical Oncology, The University of Texas MD Anderson Cancer Center, 1515 Holcombe Boulevard, Houston, Texas 77030, USA. [4] Department of Pathology, Cancer Institute and Hospital, Peking Union Medical College and Chinese Academy of Medical Sciences, Beijing 100021, People's Republic of China. [5] Beijing Genomics Institute, Shenzhen 518083, People's Republic of China. [6] Institute for Applied Cancer Science, The University of Texas MD Anderson Cancer Center, 1515 Holcombe Boulevard, Houston, Texas 77030, USA. [7] Department of Diagnostic Imaging, Cancer Institute and Hospital, Peking Union Medical College and Chinese Academy of Medical Sciences, Beijing 100021, People's Republic of China. [8] Department of Thoracic Surgical Oncology, Cancer Institute and Hospital, Peking Union Medical College and Chinese Academy of Medical Sciences, Beijing 100021, People's Republic of China. [9] Department of Bioinformatics and Computational Biology, The University of Texas MD Anderson Cancer Center, 1515 Holcombe Boulevard, Houston, Texas 77030, USA. [10] State Key Laboratory of Molecular Oncology, Department of Cellular and Molecular Biology, Cancer Institute and Hospital, Peking Union Medical College and Chinese Academy of Medical Sciences, Beijing 100021, People's Republic of China. [11] Department of Biostatistics, The University of Texas MD Anderson Cancer Center, 1515 Holcombe Boulevard, Houston, Texas 77030, USA. [12] Department of Thoracic and Cardiovascular Surgery, The University of Texas MD Anderson Cancer Center, 1515 Holcombe Boulevard, Houston, Texas 77030, USA. [13] Department of Translational Molecular Pathology, The University of Texas MD Anderson Cancer Center, 1515 Holcombe Boulevard, Houston, Texas 77030, USA. [14] Department of Biology, University of Copenhagen, DK-2200 Copenhagen, Denmark. [15] Honorary Faculty, Wellcome Trust Sanger Institute, Hinxton, Cambridgeshire CB10 1SA, UK. * These authors contributed equally to this work. ** These authors jointly supervised this work. Correspondence and requests for materials should be addressed to P.A.F. (email: AFutreal@mdanderson.org) or to Y.G. (email: yngao@cicams.ac.cn).

Lung cancer is a heterogeneous disease, with genomic and phenotypic features that differ between different patients and even between different regions of a tumour. Substantial inter-tumour heterogeneity, probably reflecting distinct genetic backgrounds and different carcinogen exposures in different patients with lung cancer, has been well documented[1,2]. On the other hand, recent studies from our group and others on non-small cell lung cancer have shown that the majority of mutations are present in all regions of a single tumour, suggesting limited intratumour heterogeneity[3,4]. Like different regions of the same tumour, multiple synchronous lung cancer (MSLC), multiple tumours arising in different areas of the lung parenchyma within a single patient, share a constitutional genetic background and exposure history. Previous studies have reported differences in certain cancer gene mutations and chromosome aberrations between different MSLCs[5–8]. The comprehensive genomic heterogeneity of MSLCs has not been well characterized but may be critical to diagnosis and appropriate treatment.

MSLCs may represent hematogenous metastases from a single primary cancer, local spread of a single primary lesion or multiple individual primary cancers. In 2007, the American College of Chest Physicians (ACCP) classified MSLCs of the same histology into satellite nodules (same lobe, no systemic metastases), multiple primary lung cancers (different lobes, no N2–N3 lymph node involvement or systemic metastases) and hematogenously spread pulmonary metastases (different lobes, N2–N3 lymph node involvement)[9]. Hematogenously spread pulmonary metastases and locally spread satellite nodules are generally believed to derive from corresponding primary tumours[10]. However, the clonal origin of multiple primary lung cancers is a subject of debate, with respect to whether they arise either independently from different progenitor cells, in line with the field cancerization concept[11], or from a single clonal event resulting in a tumour that subsequently spreads. Previous studies using targeted molecular markers obtained conflicting results[6,7,12].

To determine the genomic heterogeneity of MSLCs and assess the clonal relationships between different tumours within the same patients, we perform whole-genome sequencing (WGS) or whole-exome sequencing in combination with microarray-based comparative genomic hybridization (CGH) on 16 tumour samples (15 lung tumours (all adenocarcinomas) and one regional lymph node metastasis) from six patients with MSLCs. Five patients had satellite nodules, and one had hematogenously spread pulmonary metastasis according to the ACCP criteria (Table 1). For all 15 lung tumours, comprehensive genomic analysis revealed distinct genomic profiles, suggesting all were primary tumours.

## Results

**Somatic point mutations.** In total, 1,127 nonsynonymous coding and splice site mutations were detected (Supplementary Tables 2 and 3). Of those mutations, 956 were subjected to Sequenom's MassARRAY mass spectrometry platform or Sanger sequencing validation, and 876 (92%) were validated (Supplementary Table 4 and Supplementary Fig. 2). The remaining 171 mutations were not subjected to validation because of insufficient remaining DNA. Each of these 171 mutations was detected in only one tumour. Of the 662 nonsynonymous coding and splice site mutations called by both MuTect[13] and VarScan[14], 645 (97%) were validated.

No shared mutations were detected between different tumours from patient 2, 3 and 4 (a total of 167 mutations in six tumours), suggesting that these patients had multiple primary lung cancer (Fig. 1). In patient 1, tumour 3 (T3) and a lymph node metastasis shared 52 (26%) of 198 mutations, including a *KRAS* (p.G12V) mutation and a *STAG2* nonsense mutation (p.R305X), suggesting that the tumour metastasized to the lymph node (Figs 1 and 2). No other mutations were shared in the remaining samples from patient 1, indicating that the three lung tumours in this patient were independent primary tumours.

An *EGFR* p.L858R mutation was the only mutation shared by all three tumours of patient 6 (Figs 1 and 2). This is a known hotspot mutation and accounts for more than 40% of *EGFR* mutations reported in Asian lung adenocarcinoma patients[15]. The finding of a single prevalent hotspot mutation, however, provides limited information about tumours' independence. Indeed, comparison of *EGFR* p.L858R prevalence in this series (considering each tumour as being from a unique patient) to that

## Table 1 | Clinical characteristics and sequencing information of the six patients with multiple synchronous lung cancers.

| Patient ID | Tumour ID | Size (cm) | Location | Histology | Node staging | ACCP | Comprehensive histopathological analysis[36] | Genomic profiling | Adjunct therapy | Follow-up (months) | Recurrence | Smoking status | Sequencing/depth |
|---|---|---|---|---|---|---|---|---|---|---|---|---|---|
| Patient 1 | Pa1T1 | 1.8 | RUL | ADC | N2 | Hematogenously spread pulmonary metastases | Primary | Primary | Chemotherapy | 42 | Yes | Former smoker | WGS/37 × |
| | Pa1T2 | 1.8 | RUL | ADC | | | | | | | | | WGS/35 × |
| | Pa1T3 | 3.6 | RLL | ADC | | | | | | | | | WGS/35 × |
| | Pa1LN | 2 | Mediastinum | ADC | | | | | | | | | WGS/35 × |
| Patient 2 | Pa2T1 | 2 | LUL | ADC | N0 | Satellite nodules | Primary | Primary | No | 33 | No | Non-smoker | WES/55 × |
| | Pa2T2 | 3.1 | LUL | ADC | | | | | | | | | WES/74 × |
| Patient 3 | Pa3T1 | 3.5 | RLL | ADC | N0 | Satellite nodules | Primary | Primary | No | NA | NA | Former smoker | WES/74 × |
| | Pa3T2 | 3.5 | RLL | ADC | | | | | | | | | WES/54 × |
| Patient 4 | Pa4T1 | 0.6 | RUL | ADC | N0 | Satellite nodules | Primary | Primary | No | 38 | No | Non-smoker | WES/51 × |
| | Pa4T2 | 1 | RUL | ADC | | | | | | | | | WES/56 × |
| Patient 5 | Pa5T1 | 2 | RUL | ADC | N0 | MPLC with satellite nodules | Metastasis | Primary | No | 35 | No | Non-smoker | WES/57 × |
| | Pa5T2 | 2 | RUL | ADC | | | | | | | | | WES/56 × |
| | Pa5T3 | 0.6 | RML | ADC | | | | | | | | | WES/64 × |
| Patient 6 | Pa6T1 | 0.8 | RUL | ADC | N0 | MPLC with satellite nodules | Primary | Primary | No | 41 | No | Non-smoker | WES/52 × |
| | Pa6T2 | 0.5 | RUL | ADC | | | | | | | | | WES/58 × |
| | Pa6T3 | 1.5 | RML | ADC | | | | | | | | | WES/71 × |

ADC, adenocarcinoma; LUL, left upper lobe; RLL, right lower lobe; RML, right middle lobe; RUL, right upper lobe; WGS, whole genome sequencing; WES, whole-exome sequencing.

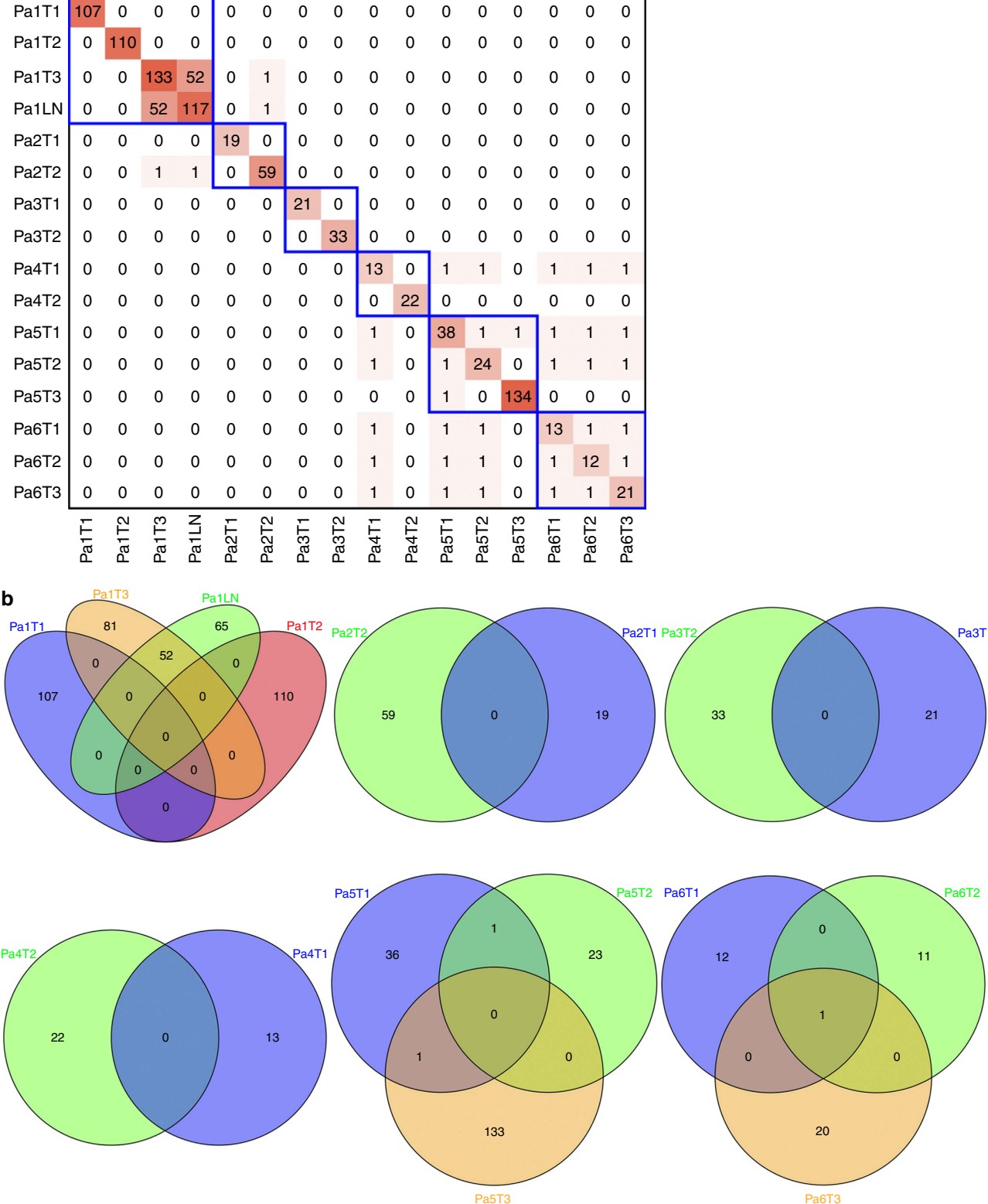

**Figure 1 | Similarity among different lesions rising from a single patient with MSLC based on somatic mutation analysis.** (**a**) Heatmap of validated mutations shared by 16 intra-thoracic adenocarcinomas of six patients with MSLC. The number of total mutations identified in each tumour (T) and the number of mutations shared by any pair of lesions are shown. Tumours from the same patients are identified by blue boxes. LN, lymph node metastasis. (**b**) Venn diagram illustrating the distributions of validated mutations in the 16 lesions. Shared mutations were defined as identical nucleotide substitutions at the same genomic coordinates.

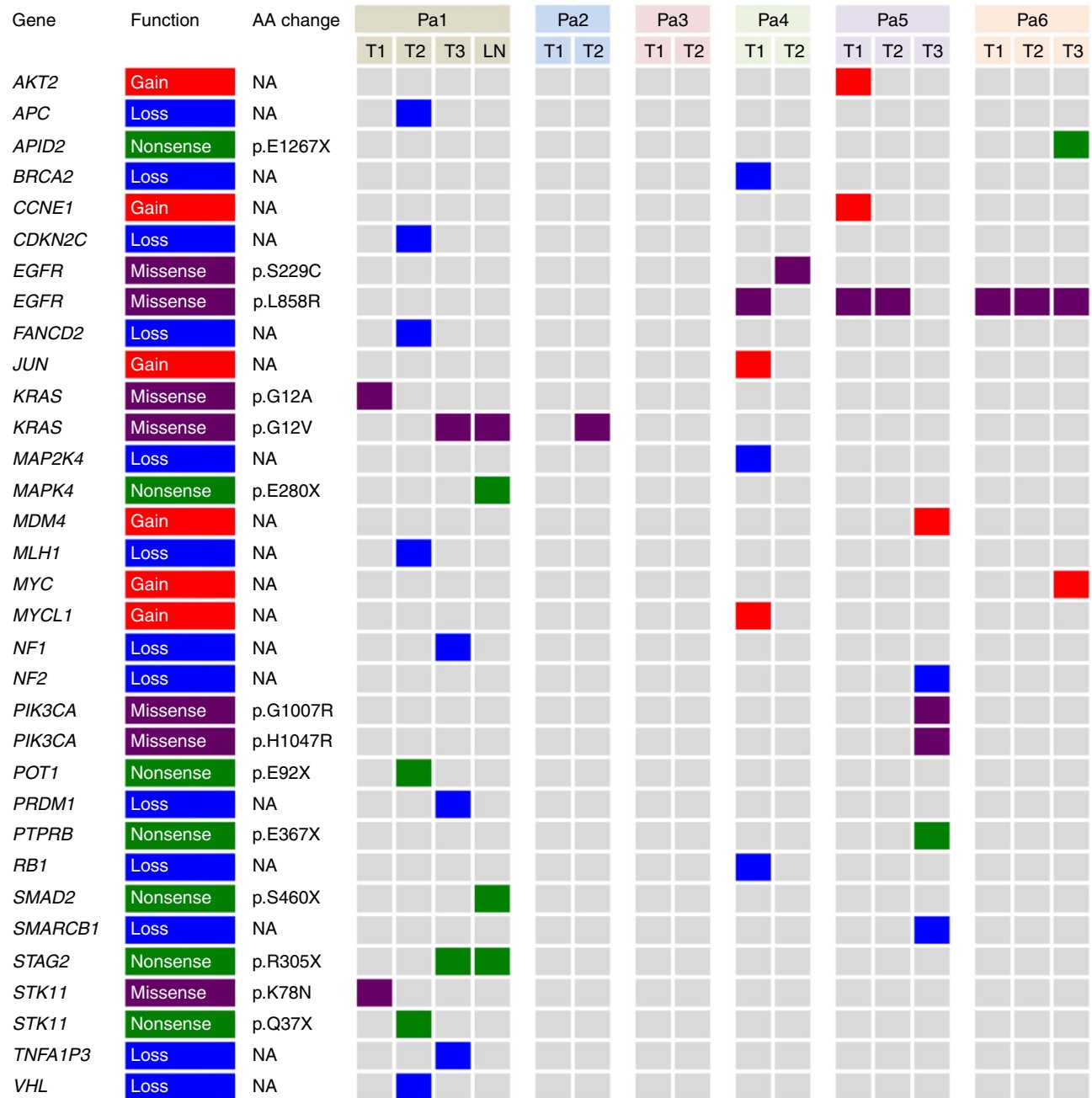

**Figure 2 | Nonsynonymous point mutations and copy number changes in known cancer genes in 16 intra-thoracic lesions of six patients with MSLC.**
Copy number changes were defined on the basis of segment $\log_2$ ratios derived from microarray-based CGH, with $\log_2$ ratios $> 0.3$ categorized as copy number gains and $\log_2$ ratios $< -0.3$ categorized as copy number losses. AA, amino acid; LN, lymph node metastasis; NA, not applicable; T, tumour number.

in a large cohort of Chinese lung adenocarcinoma patients[15] showed no enrichment in this series (6 mutations in 15 tumours (the lymph node was not included) in our study versus 111 mutations in 437 patients in the larger cohort, $P = 0.23$ by Fisher's exact test). Thus, with no other evidence of shared mutations, the data suggest that patient 6 likely had three primary tumours carrying independent *EGFR* p.L858R mutations.

In patient 5, two somatic mutations were concordant among the three tumours sequenced: an *EGFR* p.L858R mutation shared by tumours 1 and 2, and an *ARHGAP35* p.E25K mutation shared by tumours 1 and 3 (Figs 1 and 2). As with patient 6, the presence of single *EGFR* p.L858R does not provide conclusive evidence that tumours 1 and 2 of patient 5 were clonally related. *ARHGAP35*

p.E25K was the only shared mutation of the 171 coding mutations identified in tumours 1 and 3 of patient 5, which suggests two possibilities: (i) one of the two tumours was a metastasis of the other; or (ii) the two tumours arose independently, each acquiring an *ARHGAP35* p.E25K mutation during cancer development. The data from patient 1 and previous studies[4,16,17] suggest that the proportion of mutations shared between primary cancers and metastases is markedly higher than the proportion observed in patient 5 and that the observed proportion does not support a primary/metastasis relationship between tumours 1 and 3 of patient 5. To test the possibility that tumours 1 and 3 acquired *ARHGAP35* p.E25K independently, we compared our data with The Cancer Genome Atlas (TCGA) data for a series of

single primary tumours from 35 unrelated lung adenocarcinoma patients matched for tumour size and smoking status[2]. The sharing of one exonic mutation (excluding mutations in genes frequently mutated in lung adenocarcinomas) was no more likely in our series than in the TCGA cohort: 1/884 exonic mutations shared between any tumour pairs in this series versus 20/8,413 exonic mutations shared between any tumour pairs in the TCGA cohort ($P = 0.72$ by Fisher's exact test, Supplementary Table 5). Taken together, the data suggest the three tumours of patient 5 were likely independent primary tumours.

To maximize the opportunity to uncover evidence of tumour relatedness in our MSLC cohort, we expanded the mutation data set to include all validated mutations plus all mutations that were called by both VarScan[13] and MuTect[14]. Since the validation rate for nonsynonymous mutations that were called by both algorithms was 97%, an extrapolation to mutations that were not submitted to validation could be made with some confidence. Using the expanded list of 51,470 mutations, we did not identify any additional mutations shared by different tumours within the same patient (Supplementary Fig. 3), further suggesting that all 15 lung tumours of the six patients were independent primary tumours.

We then compared the exonic point mutations in the 16 MSLC lesions in our cohort and those in the 35 unrelated lung adenocarcinomas in the TCGA study[2], conservatively restricting our comparison to only T1–T2a tumours from never smokers and light smokers to make the cohorts more comparable. With the exception of tumour 3 and the associated lymph node metastasis in patient 1, each tumour shared no more than one mutation with another tumour (Supplementary Table 6). These data indicated that the MSLCs in our series were no more similar to each other than were similarly staged tumours from unrelated patients.

**Known cancer gene mutations**. We next examined the pattern of known cancer gene mutations in our series. We defined cancer gene mutations as nonsynonymous mutations identical to those previously reported in known cancer genes or truncating mutations in known tumour suppressor genes[18–23]. In total, 14 known cancer gene point mutations were identified in at least one tumour each (Fig. 2). With the exception of the *EGFR* mutations mentioned above, no known cancer gene mutations were shared between any two tumours from the same patients, suggesting that different tumours in the same patients may have been driven by different molecular events. On the other hand, three cancer genes demonstrated different mutations in different tumours of the same patients. In patient 1, tumour 3 (and the associated lymph node metastasis) harboured a *KRAS* p.G12V mutation, whereas tumour 1 had a *KRAS* p.G12A mutation. Different mutations were also observed in *STK11* in patient 1 (a missense mutation p.K78N in tumour 1 and a nonsense mutation p.Q37X in tumour 2) and in *EGFR* in patient 4 (p.L858R in tumour 1 and p.S229C in tumour 2). In addition, two different *PIK3CA* mutations, p.G1007R and p.H1047R, were found in tumour 3 of patient 5.

**Mutation spectra and mutation signature**. Our series of samples afforded the opportunity to explore mutational processes in the context of independent tumours arising on a fixed genetic background and with shared exposure. Consistent with previous studies[18,19,23,24], mutation spectra differed between smokers and non-smokers. Five of six tumours (including the metastatic lymph node) from the two former smokers (patients 1 and 3) had predominantly C>A substitutions, while eight of ten tumours from the four never smokers had largely C>T mutations. Discordant mutational spectra were observed

between same-patient tumours in all six patients, and the difference was statistically significant in patients 1, 2 and 5 (Supplementary Fig. 4), suggesting that different mutational processes were involved during the development of different tumours within the same patients. In addition, an apolipoprotein B mRNA editing enzyme, catalytic polypeptide-like (APOBEC)-mediated process[25–27] was found to contribute substantially to the mutations. However, the contribution varied between same-patient tumours. On average, 26% of the mutations showed an APOBEC-mediated pattern (C>T/G at TpCpW sites, where W is A or T). APOBEC signature enrichment was found in 15 of the 16 tumours, and the enrichment odds ratios were significant in 7 tumours of four patients (Supplementary Fig. 5).

For patient 1, WGS provided sufficient data for a more detailed mutation signature analysis[27]. Although all tumours of patient 1 showed similar mutation signatures as a group (driven mainly by the smoking-related C>A substitutions), the mutation signatures differed substantially between the individual tumours (Supplementary Fig. 6). However, tumour 3 (the largest tumour) and the associated lymph node metastasis had almost exactly the same mutation signatures, with overrepresentation of the presumptive APOBEC signature. This finding is consistent with recent evidence that APOBEC processes may be operative preferentially at a later stage in lung cancer progression[3,4]. These results suggest that, although the same dominant mutational processes may operate in different MSLC tumours during tumorigenesis (such as the mutational process driven by smoking-associated carcinogens), distinct mutational processes can be superimposed on this background in different tumours of the same patient.

**Copy number aberration**. Using microarray-based CGH, we generated somatic copy number aberration (SCNA) profiles of all tumours. In general, SCNAs were relatively few (Supplementary Fig. 7) compared with those in previous studies[1,2,18], perhaps due to the small sizes of the tumours and the fact that all patients were never smokers or former light smokers. Similar to the patterns of point mutations discussed above, the SCNA profiles of different tumours from the same patients were very different, consistent with the independent nature of these tumours (Supplementary Fig. 7). Further, amplifications and deletions of known cancer genes[21] were identified in the 15 lung tumours and the lymph node metastasis, but none was shared between different tumours of the same patients (Fig. 2).

**Indels and structural variation**. Eleven exonic small insertions/deletions (indels) were identified (none was subjected to validation because of insufficient DNA). Each indel was detected in no more than one tumour (Supplementary Tables 2 and 3), consistent with the independent, primary nature of these tumours. We were also able to evaluate genomic rearrangement profiles of the four lesions of patient 1 by WGS. With the exception of two deletions shared by tumour 3 and the associated lymph node metastasis, no common structural variants were observed between any two lesions, further supporting independent origin of the tumours investigated (Supplementary Fig. 8 and Supplementary Table 7).

**Discussion**
MSLCs have been reported to account for 0.2–8% of lung cancers[10,12,28,29], with increasing frequency of detection due to wider implementation of multislice spiral computed tomography, fluorescence endoscopy and positron emission tomography[30,31]. In this study, we performed comprehensive genomic profiling of 15 multiple synchronous lung

adenocarcinomas and one lymph node metastasis from 6 patients. Despite shared genetic background and exposure history, all same-patient lung tumours had distinct genomic profiles, including somatic point mutations, copy number aberrations, chromosomal structural variations and even mutational spectra. Tumours of the same individuals were no more similar to each other than lung adenocarcinomas of different patients (TCGA data) matched for tumour size and patient smoking status. These data provide evidence that multiple mutational processes may be in play during the development of independent lung tumours within the same individual subjected to common exposures on the same constitutional genetic background.

In addition, several cancer genes had different mutations in different tumours within the same patients. This finding is reminiscent of intratumour heterogeneity observed for clear cell renal cell carcinomas[32], with different mutations in the same cancer gene found in different subclones of the primary tumour suggesting convergent selection. Although our sample size was small, these results suggest that even in the context of identical genetic background and environmental exposure, the development of multiple primary lung adenocarcinomas can be driven by distinct molecular events in different tumours, with possible selection constraints around certain genes/pathways that are critical for carcinogenesis in specific patients.

MSLCs could be multiple primary tumours that are potentially curable or intrapulmonary metastases that could be taken as an indication of unresectable disease. Many attempts have been made to distinguish these clinical entities. Martini–Melamed criteria[33] and ACCP guidelines[9] are widely adapted clinical guidelines although they are rather empirical with little supporting molecular evidence. In this study, we profiled 6 patients who were all defined as clinically metastatic (intrapulmonary metastasis or satellite nodules) by ACCP guidelines (therefore, may be otherwise excluded from curative therapies). However, genomic profiling suggested that all 6 patients in fact had multiple primary tumours. With a minimum follow-up of 33 months post surgery, none of the patients with satellite nodules has relapsed, while the patient with hematogenously spread pulmonary metastasis (patient 1) relapsed 42 months after surgery. Previous studies have demonstrated a slightly shorter survival in patients with satellite nodules compared with patients without satellite nodules matched for primary tumour size, lymph node and metastatic stage[34]. Our data suggest that a substantial proportion of tumours categorized as hematogenously spread pulmonary metastases and satellite nodules may instead be multiple primary tumours. To improve the diagnostic accuracy of MSLCs, pioneering studies led by Travis *et al.* have investigated comprehensive histologic assessment and have shown promising results[8,35,36]. Using similar approach, we were able to accurately identify 5 of the 6 patients (Supplementary Table 1) confirming that comprehensive histologic assessment is highly valuable to distinguish multiple primary tumours from intrapulmonary metastases in majority of patients. However, morphology is presumably controlled by complex molecular mechanisms, of which our knowledge is rudimentary. Tumours can change their histologic appearance from one to another. Thus, the morphology similarity between different tumours could be suggestive but not conclusive. On the other hand, as shown in this study, with the caveat of the small sample size fully acknowledged, multiple primary tumours have distinct genomic profiles, while metastatic lesions usually retain a significant fraction of genomic aberrations from the founding primary tumours[16,17,37]. Therefore, comprehensive genomic profiling at the exome level, can provide pivotal information to clinical and histologic assessment to accurately distinguish multiple primary lung cancers from intrapulmonary metastases. Application of genomic profiling in the clinical setting of staging

patients with MSLCs should be explored in a larger cohort to confirm the utility suggested here. If corroborated, genomic profiling may prove an important component of a more precise approach in managing patients presenting with MSLC.

## Methods

**Patients.** Surgical specimens and peripheral blood samples were collected from six patients who were diagnosed with pathologically confirmed multiple synchronous lung adenocarcinomas, with two or three tumours in the same lung, and treated at the Cancer Institute and Hospital, Chinese Academy of Medical Sciences, Beijing, China. Tumour sizes ranged from 0.5 to 3.6 cm according to pathology reports. All patients were free of extrathoracic metastasis. Patient 1 had metastasis to a mediastinal lymph node, but no other patient had lymph node involvement. None of the patients had pre-operative chemotherapy or radiation therapy. Four patients were never smokers, and two were former smokers. The patients' clinical characteristics are listed in Table 1, and tumour characteristics are shown in Supplementary Table 1 and Supplementary Fig. 1. The collection and analysis of patient samples were approved by the Ethics Committee of the Cancer Institute and Hospital, Chinese Academy of Medical Sciences. Informed consent was obtained from all patients.

**Sample collection and processing.** After resection, ten 10 μm fresh frozen sections for each tumour sample or ten 10 μm formalin-fixed paraffin-embedded sections for the regional lymph node metastasis were collected. Haematoxylin–eosin-stained slides (Supplementary Fig. 1B) were reviewed by experienced lung cancer pathologists to determine the histomorphological subtype and the proportion of malignant cells relative to nonmalignant stromal (inflammatory, vascular and fibroblast) cells. In addition, tumour cell viability was addressed by examining the presence of necrosis in the tissues. Tumour cells were enriched by having a pathologist scrape tumour tissues from each slide. Genomic DNA was then extracted from all samples, and matched peripheral blood leukocytes were used as a germline DNA control.

**Whole-genome sequencing.** Genomic DNAs were fragmented into 500-bp segments by using the Covaris (Woburn, MA) E210 instrument. The double-stranded DNA fragments consisted of 3′ or 5′ overhangs. T4 DNA polymerase and Klenow enzyme (Invitrogen, Life Technologies, Grand Island, NY) were then used to convert the overhangs into blunt ends. An A base was added to the 3′-end of the blunt phosphorylated DNA fragments, which was ligated with adapters on both ends. The correctly ligated products were purified by agarose gel electrophoresis and then with the QIAquick Gel Extraction Kit (Qiagen, Valencia, CA). DNA fragments with adapter molecules on both ends were selected and amplified. After PCR using primers that anneal to the ends of the adapters, the products were checked and purified by agarose gel electrophoresis and sequenced using the HiSeq 2000 system (Illumina, San Diego, CA). The average sequencing depth was $35 \times$ per sample (range, $35 \times$ to $37 \times$; s.d., $0.6 \times$).

**Whole-exome sequencing.** Genomic DNAs from patients 2 to 6 were sheared into fragments with peaks of 150–200 bp, and then adapters were ligated to both ends. The adapter-ligated templates were purified with Agencourt AMPure SPRI beads (Beckman Coulter, Inc., Brea, CA), and fragments with an insert size of ∼200 bp were excised. Extracted DNA was amplified by ligation-mediated PCR, purified and hybridized to the SureSelect biotinylated RNA library (Agilent Technologies, Santa Clara, CA) for enrichment according to the manufacturer's instructions. Paired-end multiplex sequencing of samples was performed with the IlluminaHiSeq 2000 System. The average sequencing depth was $62 \times$ per sample (range, $51 \times$ to $74 \times$; s.d., $9 \times$).

**Sequence alignment and variant calling.** Paired-end reads in FastQ format generated by the Illumina pipeline were aligned to the reference human genome (UCSC Genome Browser, version hg19) using the Burrows-Wheeler Aligner with default settings[38], except for a seed length of 40, a maximum edit distance of 3 and a maximum edit distance in the seed of 2. Aligned reads were further processed according to the Genome Analysis Toolkit (GATK) Best Practices (www.broadinstitute.org/gatk/guide/best-practices.php) for duplicate removal, indel realignment and base recalibration.

Both VarScan[13] version 2.2.5 and MuTect[14] were used to detect potential single-nucleotide variations. For VarScan, in addition to the built-in filters, the following filtering criteria were applied: (i) coverage $\geq 10$ in germline DNA and $\geq 4$ in tumour DNA; (ii) variant frequency $\geq 10\%$; and (iii) $P < 0.0001$ for calling a somatic site. For MuTect, in addition to the build-in filters, the following filtering criteria were applied: (i) total read count in tumour DNA $\geq 15$; (ii) total read count in germline DNA $\geq 6$; (iii) presence of variant on both strands; (iv) variant allele frequency in tumour DNA $\geq 10\%$; (v) variant allele frequency in germline DNA $= 0$; and (vi) removal of variants in positions listed in the dbSNP129 database (www.ncbi.nlm.nih.gov/SNP/). Single-nucleotide variants called by either method were used for further analysis (Supplementary Tables 2–4).

The GATK SomaticIndelDetector[39,40] was used to detect potential somatic indels. The following filtering criteria were applied: (i) read depth > 5 in both tumour and normal samples; (ii) average mismatch rate < 0.5 in both normal and mutant alleles; (iii) average mapping quality > 20 in both normal alleles and mutant alleles in a tumour; and (iv) median indel offsets from the end of the reads > 5 bp (Supplementary Tables 2 and 3).

For samples from patient 1, which were subjected to WGS, the CREST (Clipping Reveals Structure) algorithm[41] was implemented to identify potential structural variants. Only breakpoint pairs with at least five supporting clipped reads spanning the breakpoints and at least one supporting clipped read for each end were selected for validation and further analysis.

**Somatic variant validation.** Nonsynonymous coding and splice site mutations called by either MuTect or VarScan were subjected to mass spectrometry or Sanger sequencing for validation when adequate DNA was available. Mass spectrometry was performed first with the MassArray platform (Sequenom, San Diego, CA) as previously described[42]. For mutations for which Sequenom software failed to design primers for amplification, Sanger sequencing was applied for validation. Sanger sequencing was also used to validate structural variants.

**Detection of SCNAs by microarray-based CGH.** We performed SCNA analysis using the Human Genome CGH Microarray Kit 244A (Agilent Technologies) with 8.9 kb overall median probe spacing, according to the ULS Labeling Protocol for Agilent Oligonucleotide Array-Based CGH for Genomic DNA Analysis (version 3.4, July 2012). After scanning with the Agilent Scanner System, the data in each slide were extracted with Feature Extraction 12.0 (Agilent Technologies) for further analysis. The extracted data were subjected to locally weighted scatterplot smoothing to remove potential intensity and/or GC content bias before calculating the $\log_2$ copy number ratios in reference to the matching normal. $\log_2$ ratios for each tumour sample were then segmented by applying the circular binary segmentation algorithm[43]. Copy number gain was defined as segmented $\log_2$ ratio > 0.3, and copy number loss was defined as $\log_2$ ratio < − 0.3. Cancer genes known to be affected by amplification or deletion (http://cancer.sanger.ac.uk/cancergenome/projects/census/) were also screened using these thresholds. Manual inspection was applied to review all segments containing candidate genes in each tumour region to make amplification and deletion calls. We also assessed the clonal relationship between different tumours based on the likelihood ratio against a background reference distribution as previously described[7]. Briefly, copy number segmentation data were partitioned into overlapping regions across samples. Pair-wise correlations were calculated for all potential pairs between as well as within tumours. Inter-tumour correlations were plotted as the background distribution, and intratumour correlations were plotted in shade. Finally, adjacent segments were merged and annotated with recurrent copy number changes of lung adenocarcinomas referenced in the TCGA Copy Number Portal (http://www.broadinstitute.org/tcga/gistic/browseGisticByTissue;jsessionid=08F9235B734370DB93AF3A4A33D86DB9). Segments overlapping with ≥ 50% of the recurrently amplified or lost regions were classified as gains or losses, respectively.

**APOBEC mutation signature analysis.** APOBEC mutation signatures were analysed as previously described[3]. In brief, APOBEC signature enrichment $E_{TCW}$ in relation to the strength of mutagenesis at the TCW motif (where W is either A or T) was calculated as in equation (1):

$$E_{TCT} = \frac{\text{mutations}_{TCW} \times \text{context}_{CorG}}{\text{mutations}_{CorG} \times \text{context}_{TCW}} \tag{1}$$

where mutations$_{TCW}$ is the number of mutated cytosines (and guanines) in a TCW (or WGA) motif, mutations$_{CorG}$ is the total number of mutated cytosines (or guanines), context$_{TCW}$ is the total number of TCW (or WGA) motifs within a 41-nucleotide region centred on the mutated cytosines (and guanines) and context$_{CorG}$ is the total number of cytosines (or guanines) within the 41-nucleotide region centred on the mutated cytosines (or guanines). Only the following substitutions were included in the analysis: TCW to TTW or TGW and WGA to WAA or WCA. Overrepresentation of the APOBEC mutation signature was determined using a two-sided Fisher's exact test comparing the ratio of cytosine-to-thymine or cytosine-to-guanine substitutions with guanine-to-adenine or guanine-to-cytosine substitutions that occurred in and out of the APOBEC target motif (TCW or WGA) to an analogous ratio for all cytosines and guanines inside and outside the TCW or WGA motif within the 41-nucleotide region centred on the mutated cytosine or guanine.

**Statistical analyses.** Analysis of variance was used to assess the association between mutation burden and the gender or smoking status of each patient. The Pearson product–moment correlation test was used to assess the association between mutation burden and each patient's age or tumour size. The Fisher's exact test was used to assess the significance of differences in mutation spectra between different tumours, and the Pearson product–moment correlation analysis was used to assess the correlation between the mutation spectra of different tumours. The Fisher's exact test was also used to compare the incidences of *EGFR* p.L858R in our cohort and the Chinese lung adenocarcinoma cohort[15]. To determine the correlation of SCNA profiles between different tumours from the same patients, we

processed segmented data using the Bioconductor CNTools software package to generate a gene-by-tumour-region copy number matrix. Correlations between different tumours were then calculated to obtain correlation coefficients.

**Data availability.** Whole-genome and -exome sequencing data have been deposited at the European Genome-phenome Archive (www.ebi.ac.uk/ega/), which is hosted by the European Bioinformatics Institute (accession number: EGAS00001001572). The aCGH data have been deposited in the GEO database under accession code GSE86607. All other data are included within the Article or Supplementary Information or available from the authors on request.

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

## Acknowledgements

This study was supported by the National Basic Research Program of China (2014CB542002 to Y.G.), National High-tech R&D Program of China (2012AA02A502 and 2006AA02A401 to Y.G.), the Fundamental Research Funds for the State Key Laboratory (SKL-2013-05 to Y.G.), the Capital Health Research and Development Special Fund of China (2011-4002-01 to D.L.), the National Natural Science Foundation of China (81472743 to D.L.), the Cancer Prevention and Research Institute of Texas (RP160668 to I.W., P.A.F. and Jianjun Zhang and R120501 to P.A.F.), The University of Texas System STAR Award (PS100149 to P.A.F.), the Welch Foundation's Robert A. Welch Distinguished University Chair Award (G-0040 to P.A.F.) and the C.G. Johnson Advanced Scholar Program (to Jianjun Zhang), the MD Anderson Moon Shot Program (to Jianjun Zhang), MD Anderson Physician Scientist Program (to Jianjun Zhang), the Khalifa Scholar Award (to Jianjun Zhang) and the Conquer Cancer Foundation Young Investigator Award (to Jianjun Zhang). We thank Dr G. Draetta for constructive discussions. The study funders had no role in the design of the study; the collection, analysis or interpretation of the data; the writing of the manuscript; or the decision to submit the manuscript for publication.

## Author contributions

As senior principal investigators, Y.G., P.A.F., D.L., X.L. and J.W. designed and coordinated the study; Y.L., Jianjun Zhang, P.A.F. and Y.G. were primarily responsible for data analysis, data interpretation and the writing of the manuscript; L.Y., C.Z. and X.L. obtained patient consent and collected tissue samples; S. Zheng, N.L., Y.X. and D.L. performed pathology reviews; N.W., L. Zhang, K.L. and L. Zhou performed radiology reviews; L. Li (Chinese Academy of Medical Sciences), H. Chen, N.H., W. Chen and S. Zhang collected tissue samples, prepared DNA samples and carried out microarray-based CGH; L.C., W. Cai, L. Li (Beijing Genomics Institute), M.S. and H.Y. performed DNA sequencing; G.Y. and Jianhua Zhang had overall responsibility for mutational analysis and data analysis; X.M., S. Seth and X.S. ran the data mutational analysis pipeline; Jiexin Zhang and J.J.L. performed the statistical analyses; H. Cheung, S.C., N.X., A.C., J.F., C.B., C.-W.C., W.N.W., J.V.H., W.K.H., S. Swisher and I.I.W. participated in data interpretation, analysis of clinicopathological correlations and manuscript writing.

## Additional information

**Competing financial interests:** The authors declare no competing financial interests.

