## [Peer Review File · Nature Communications]

Reviewers' comments:

Reviewer #1 (Remarks to the Author):

A. Summary: Until now, it has been mysterious whether independent primary lung cancers are truly independent from one another or are clonally related to one another. Using systematic genomics approaches, the manuscript by Liu et al. answers this question definitively: multiple synchronous lung cancers are genomically distinct and therefore are almost certain to arise independently from one another.

In this study, Liu et al. analyzed exome and genome sequences from multiple synchronous lung cancers. They find that these cancers are genomically distinct; for example, they exhibit distinct and independent mutations of EGFR, KRAS, and STK11. In another interesting case, they find tumors from a patient that all share a common EGFR mutation, L858R, but show that these tumors are nevertheless genomically distinct because of independent mutations at other positions.

B. The paper is highly original and interesting.

C. The methods and quality appear very high. The figures are visually boring which may have caused difficulty if this paper was previously submitted to another journal, but I think the figures accurately and clearly communicate the data.

D. Statistics are appropriate in my opinion but a more expert statistical reviewer might have more suggestions.

E. Conclusions: Robust, reliable, and fascinating.

F. Suggested improvements: none.

G. References: appropriate in my opinion.

H. Clarity and context: Paper is clearly written and easy to follow.

Reviewer #2 (Remarks to the Author):

The authors Liu et al. here report on 6 patients presenting with 15 intrapulmonary adenocarcinoma, 5 with satellites nodules and one with hematogeneously spread (lymph node)metastasis. In these 6 patients, they use frozen tissue - extracted DNA from enriched tumor areas in frozen sections to perform comprehensive genomic analysis (WES/WGS) in order to compare the different tumors of the same patients for non-synonymous mutations, structural genes alterations (indel...) and CGH for gene copy number alterations, as well as APOBEC signatures. They thus demonstrate that multiple tumors from the same patients display distinct genomic profiles and may be driven by distinct molecular events, highly suggesting that they derive from distinct clonal proliferations.

Despite all patients would have been considered on AJCC 2003 staging system (and Martini Melamed recommendations) as having intrapulmonary metastases, 5 with satellites nodules and one with hematogeneously spread metastasis, all were found to be independent primaries, not clonally related tumors in the same patients.

- These findings are of sufficient originality and interest for the oncologist and pathologist community. Previous studies were not as comprehensive mostly including a few driver oncogenes events and CGH data. Please cite and discuss Girard et al: AM. J. SURG. PATHOL. 2009). It rises valid clinical question in the context of increasing early radiological lung cancer detection, leading to discovery of more than one tumor nodule in the same patient.

- This series of cases is rather short : what where the limitations of cases inclusion ? the series is quite limited , likely due to requirement of frozen tissues in order to extract good quality DNAs ?
- All patients included were prone to be considered clinically as metastatic (intrapulmonary metastases) . Could the authors be more clear on their patients criteria selection ? and therefore on their clinicopathologic question and rational ?
- The low number of cases implies that the comprehensive assessment of their histologic subtypes of adenocarcinoma is feasible .This is of primary importance due to the current updating of the 7th staging and a proposal of 8th edition which implies that few features are sufficiently reliable to establish that 2 pulmonary foci of cancer are separate primary tumor, rather than metastatic from one other : 1- Different histological (major) type (here all tumors are adenocarcinoma); 2- difference by comprehensive histologic assessment (authors should make a comprehensive histological review of the 15 tumors according to the WHO 2015 adenocarcinoma classification into predominant and non- predominant patterns , and evaluate if there is radiological GGO /histological lepidic components ,which is a clue for primary nature of a tumor (since all tumor are small size) ; 3- non -matching breakpoints by DNA sequencing (which is done here and lead to conclude to different rearrangements and structural alterations ?). It is recommended that a multidisciplinary concertation , the only standard being clinical outcome , establishes the status of independent multiple primaries versus metastases (J T O 2016 F. Detterbeck et al . 2016 March 1 epub ahead) . It is so highly expected that a level of concordance between morphology(comprehensive histopathological analysis according to WHO2015 classification of lung tumors and genomic profiles are of help in taking the decision of synchronous primaries established in this short series.
- Many attempts have been made or are ungoing to predict the classification as multiple primaries against intra pulmonary metastases , according to differences or similarities of tumors on histological ground , based on the concept that identical tumors with identical pattern distribution are proposed to be metastatic from one primary but this is only suggestive and also subject of missclassifications.(50% of second primaries are of the same major histological type than the first !)
- In the discussion the authors discuss the intra-tumoral heterogeneity of Clear cell renal cancer (Gerlinger ref.31) . A discussion should include a previous comprehensive genomic comparison of primary NSCLC and their intra-lung or distant metastasis , showing a clear similarity of driver genomic alterations of more than 90% in 15 patients withtheir paired metastatic tumor , despite 40% variations in otherwise passenger mutations . Theses patients were all clinically metastatic (Vignot et al . J.C.O. 2013,31(17)) showing at least the high conservative molecular genomic profiles in the lung primary and metastases ,in contrast with renal carcinoma .
- The abstract is appropriate pending addition of the criteria of selection of the patients and the standard applying for the consideration of primary synchronous tumors rather than intra-pulmonary metastases.
- The discussion should include missing reference
- The conclusion may announce the added value of the present study for consideration of requirement of this type of molecular study ,in addition to the histopathological assessment of differences and similarities which are more subjective but more compli cated and coastly and less feasible on the clinical ground of staging of patients with multiple synchronous lung cancers.

Point-point response to reviewers' suggestions

Reviewer #1 (Remarks to the Author):

A. Summary: Until now, it has been mysterious whether independent primary lung cancers are truly independent from one another or are clonally related to one another. Using systematic genomics approaches, the manuscript by Liu et al. answers this question definitively: multiple synchronous lung cancers are genomically distinct and therefore are almost certain to arise independently from one another.

In this study, Liu et al. analyzed exome and genome sequences from multiple synchronous lung cancers. They find that these cancers are genomically distinct; for example, they exhibit distinct and independent mutations of EGFR, KRAS, and STK11. In another interesting case, they find tumors from a patient that all share a common EGFR mutation, L858R, but show that these tumors are nevertheless genomically distinct because of independent mutations at other positions.

B. The paper is highly original and interesting.

C. The methods and quality appear very high. The figures are visually boring which may have caused difficulty if this paper was previously submitted to another journal, but I think the figures accurately and clearly communicate the data.

D. Statistics are appropriate in my opinion but a more expert statistical reviewer might have more suggestions.

E. Conclusions: Robust, reliable, and fascinating.

F. Suggested improvements: none.

G. References: appropriate in my opinion.

H. Clarity and context: Paper is clearly written and easy to follow.

Authors: We appreciate the favorable comments and the reviewer's interpretation that our manuscript has significant value.

Reviewer #2 (Remarks to the Author):

The authors Liu et al. here report on 6 patients presenting with 15 intrapulmonary adenocarcinoma, 5 with satellites nodules and one with hematogeneously spread (lymph node) metastasis. In these 6 patients, they use frozen tissue – extracted DNA from enriched tumor areas in frozen sections to perform comprehensive genomic analysis (WES/WGS) in order to compare the different tumors of the same patients for non- synonymous mutations, structural genes alterations (indel...) and CGH for gene copy number alterations, as well as APOBEC signatures. They thus demonstrate that multiple tumors from the same patients display distinct genomic profiles and may be driven by distinct molecular events, highly suggesting that they derive from distinct clonal proliferations. Despite all patients would have been considered on AJCC 2003 staging system (and Martini Melamed recommendations) as having intrapulmonary metastases, 5 with satellites nodules and one with hematogeneously spread metastasis), all were found to be independent primaries, not clonally related tumors in the same patients.

- These findings are of sufficient originality and interest for the oncologist and pathologist community. Previous studies were not as comprehensive mostly including a few driver oncogenes events and CGH data. Please cite and discuss Girard et al: AM. J. SURG. PATHOL. 2009). It rises valid clinical question in the context of increasing early radiological lung cancer detection, leading to discovery of more than one tumor nodule in the same patient.

Authors: We appreciate the favorable comments and we agree with reviewer's suggestion and have cited and discussed relevant studies in the revised version.

- This series of cases is rather short: what were the limitations of cases inclusion? The series is quite limited, likely due to requirement of frozen tissues in order to extract good quality DNAs?

Authors: Just as the reviewer suspected, the sample size was limited by the quantity and quality of samples. As a proof-for-principle study, we only used frozen tumor tissues with highly enriched malignant cells and matched peripheral blood leukocytes available as germ line DNA control to get high confident somatic aberrations.

- All patients included were prone to be considered clinically as metastatic (intrapulmonary metastases). Could the authors be more clear on their patients criteria selection? and therefore on their clinicopathologic question and rationale?

Authors: We agree with reviewer that all patients in this study were prone to be considered clinically metastatic. We chose a cohort of such patients because we think these patients are clinically more challenging. In a major academic institute, patients presenting with multiple lung cancers are often managed by a multidisciplinary team to make the treatment plan. Curative surgery or radiation is often offered to many patients, including patients with intrapulmonary metastases by ACCP guidelines or Martini-Melamed criteria (as for the patients in our study). However, in the community, these criteria are still widely used although they are rather empirical. As a result, patients defined as metastatic are often excluded from potentially curative treatments. The larger proportion of suspected/diagnosed metastatic disease cases here weights toward perhaps the more immediate clinical dilemma.

- The low number of cases implies that the comprehensive assessment of their histologic subtypes of adenocarcinoma is feasible. This is of primary importance due to the current updating of the 7th staging and a proposal of 8th edition which implies that few features are sufficiently reliable to establish that 2 pulmonary foci of cancer are separate primary tumor, rather than metastatic from one other: 1- Different histological (major) type (here all tumors are adenocarcinoma); 2-difference by comprehensive histologic assessment (authors should make a comprehensive histological review of the 15 tumors according to the WHO 2015 adenocarcinoma classification into predominant and non-predominant patterns , and evaluate if there is radiological GGO/histological lepidic components, which is a clue for primary nature of a tumor (since all tumor are small size); 3- non-matching breakpoints by DNA sequencing (which is done here and lead to conclude to different rearrangements and structural alterations?). It is recommended that a multidisciplinary concertation, the only standard being clinical outcome, establishes the status of independent multiple primaries versus metastases (JTO 2016 F. Detterbeck et al. 2016 March 1 epub ahead). It is so highly expected that a level of concordance between morphology (comprehensive histopathological analysis according to WHO2015 classification of lung tumors and genomic profiles are of help in taking the decision of synchronous primaries established in this short series.

Authors: As discussed above, we totally agree with the reviewer that ideally, a multidisciplinary approach including histological, radiological and molecular assessment should be applied to the diagnosis of multiple lung cancers. We did perform a comprehensive pathological and radiological assessment for each patient according to WHO2015 classification of lung tumors as shown in Supplementary Table 1. We thank the reviewer's constructive suggestion on the newly published study. We have now updated Table 1 and Supplementary Table 1 with information on histological subtypes and radiological characteristics. As the reviewer suspected, genomic profiling is consistent with comprehensive histological assessment in 5 out of 6 patients, suggesting that comprehensive pathological assessment is highly valuable to distinguish multiple primary tumors from intrapulmonary metastases in majority of patients. We cited relevant studies and discussed about this topic in more details in the revised version.

- Many attempts have been made or are ongoing to predict the classification as multiple primaries against intrapulmonary metastases, according to differences or similarities of tumors on histological ground, based on the concept that identical tumors with identical pattern distribution are proposed to be metastatic from one primary but this is only suggestive and also subject of misclassifications. (50% of second primaries are of the same major histological type than the first!)

Authors: We totally agree with the reviewer. The morphology similarity between different lesions could be suggestive but not conclusive. Both could be true when the same major histopathology subtype was shared between lesions. As shown in the revised Supplementary Table 1, the lymph node metastasis from patient 1, shared the same major subtype with its source tumor Pa1T3, and they do demonstrate very similar genomic profiles - being consistent between morphological and genomic assessment. On the other hand, although T1 and T2 from patient 5 also shared the major histopathology subtype, the genomic profiles from these tumors are totally different. These data suggest that genomic profiling may add additional information to comprehensive histological assessment.

- In the discussion the authors discuss the intra-tumoral heterogeneity of Clear cell renal cancer (Gerlinger ref.31). A discussion should include a previous comprehensive genomic comparison of primary NSCLC and their intra-lung or distant metastasis, showing a clear similarity of driver

genomic alterations of more than 90% in 15 patients with their paired metastatic tumor, despite 40% variations in otherwise passenger mutations. These patients were all clinically metastatic (Vignot et al. J.C.O. 2013,31(17)) showing at least the high conservative molecular genomic profiles in the lung primary and metastases in contrast with renal carcinoma.

Authors: We agree with the reviewer and we believe the discussion makes the argument more convincing. We cited and discussed about this topic in more details in the revised manuscript.

- The abstract is appropriate pending addition of the criteria of selection of the patients and the standard applying for the consideration of primary synchronous tumors rather than intra-pulmonary metastases.

Authors: We have modified the abstract accordingly.

- The discussion should include missing reference

Authors: We have cited appropriate references as suggested in the revised manuscript.

- The conclusion may announce the added value of the present study for consideration of requirement of this type of molecular study in addition to the histopathological assessment of differences and similarities which are more subjective but more complicated and costly and less feasible on the clinical ground of staging of patients with multiple synchronous lung cancers.

Authors: We have discussed about the utility and feasibility in more details in the revised manuscript.

REVIEWERS' COMMENTS:

Reviewer #2 (Remarks to the Author):

A Summary of the key results is optimal on the revised manuscript corresponding to this reviewer expectations .

B As already mentionned it is an original manuscript using comprehensive molecular analysis to demonstrate that multiple synchronous tumor are potentially synchronous primaries rather than metastatic as inferred from AJCC and Martini Melamed previous recommendations

The conclusions driven by the data are robust

The authors have correctly amended their manuscript according to reviewer questions : The radiological and comprehensive histopathological review have been added on fig sup. fig 1 and figure 1

The missing referenced have been added in optimal manner and enriched the paper

The introduction and conclusions have now clearly exposed the clinicopathological rationale and the clinical need to correctly classify patients with MSLC . The added value of molecular comprehensive approach to comprehensive histopathological review appears more clearly ,as well as the requirement for a multidisciplinary approach to solve the problematic question of primaries against metastases

Point-point response to reviewers' suggestions

REVIEWERS' COMMENTS:

Reviewer #2 (Remarks to the Author):

A Summary of the key results is optimal on the revised manuscript corresponding to this reviewer expectations.

B As already mentioned it is an original manuscript using comprehensive molecular analysis to demonstrate that multiple synchronous tumor are potentially synchronous primaries rather than metastatic as inferred from AJCC and Martini Melamed previous recommendations.

The occlusions driven by the data are robust.

The authors have correctly amended their manuscript according to reviewer questions: The radiological and comprehensive histopathological review have been added on fig sup. fig 1 and figure 1.

The missing referenced have been added in optimal manner and enriched the paper The introduction and conclusions have now clearly exposed the clinicopathological rational and the clinical need to correctly classify patients with MSLC. The added value of molecular comprehensive approach to comprehensive histopathological review appears more clearly as well as the requirement for a multidisciplinary approach to solve the problematic question of primaries against metastases.

Authors: We appreciate the favorable comments and the reviewer's interpretation that our manuscript has significant value.